# Immune Sensitization to *Mycobacterium tuberculosis* Among Young Children with and without Tuberculosis

**DOI:** 10.3390/pathogens14090924

**Published:** 2025-09-12

**Authors:** Jesús Gutierrez, LaShaunda L. Malone, Mitchka Mohammadi, John Mukisa, Michael Atuhairwe, Simon Peter G. Mwesigwa, Salome Athieno, Sharon Buwule, Faith Ameda, Sophie Nalukwago, Ezekiel Mupere, Catherine M. Stein, Christina L. Lancioni

**Affiliations:** 1Department of Population and Quantitative Science, Case Western Reserve School of Medicine, Cleveland, OH 44106, USA; 2Uganda-Case Western Reserve University Research Collaboration, Kampala 25601, Uganda; 3School of Medicine, Oregon Health and Science University, Portland, OR 97239, USA; 4Department of Radiology and Radiotherapy, College of Health Sciences, Makerere University, Kampala 7072, Uganda; 5Joint Clinical Research Centre, Kampala 10005, Uganda; 6Department of Paediatrics, School of Medicine College of Health Sciences, Makerere University, Kampala 7072, Uganda; 7Department of Pediatrics, Oregon Health and Science University, Portland, OR 97239, USA

**Keywords:** *Mycobacterium tuberculosis*, immune sensitization, household contact, tuberculin skin test, interferon gamma release essay, QuantiFERON Gold Plus

## Abstract

Identification of young children with Mycobacterium tuberculosis (Mtb) infection is critical to curb pediatric morbidity and mortality. The optimal test to identify young children with Mtb infection remains controversial. Using a tuberculosis (TB) household contact (HHC) study design among 130 Ugandan children less than 5 years of age with Mtb exposure, this study was conducted to determine the following: (1) the prevalence of Mtb immune sensitization in young children heavily exposed to TB using both the tuberculin skin test (TST) and QuantiFERON Gold Plus (QFT-Plus) interferon gamma release assay, and to examine the concordance of these two tests; and (2) the diagnostic accuracy of TST and QFT-plus for confirmed and unconfirmed TB in young children. Prevalence of Mtb immune sensitization was determined using TST at both 5 mm and 10 mm thresholds for positivity; manufacturer’s thresholds were utilized to establish QFT-Plus positivity. Concordance analysis between TST and QFT-Plus results was performed, including correlation between QFT-Plus tube TB.1 and tube TB.2. The sensitivity and specificity of TST and QFT-Plus for confirmed and unconfirmed TB was determined, and a logistic regression model was utilized to estimate the odds of TB. A 5 mm TST threshold identified the most children with Mtb sensitization (49.2%) and had moderate agreement with QFT-Plus (Cohen’s Kappa 0.59). The odds of TB were two times higher among children with a positive TST using a 5 mm threshold. Concordance between 10 mm TST threshold and QFT-Plus was substantial (Cohen’s Kappa 0.65), with higher concordance observed among older children (2–5 years). The QFT-Plus tube TB.1 and tube TB.2 results were highly correlated. Positive TST using a 5 mm threshold demonstrated the highest sensitivity for TB (60%), whereas QFT-Plus testing demonstrated the highest specificity (72%). Overall, our findings support that among a population of young, BCG-vaccinated children with heavy household exposure to TB, the TST using a 5 mm threshold identified more children with evidence of Mtb immune sensitization, and children with TB disease, than the QFT-Plus. These findings are highly relevant for children who are TB HHCs in endemic settings, and most at risk for TB following an exposure. We recommend that TST testing continue to be performed to assess for Mtb sensitization in young, TB-exposed children in TB-endemic settings to both prioritize provision of preventive therapy and to aide in diagnosis of pediatric TB.

## 1. Introduction

Over 1 million children develop tuberculosis (TB) annually, with close to 250,000 pediatric deaths attributed to TB every year [1]. Due to difficulties in diagnosing TB in children and limitations in public health resources in many TB-endemic settings, most children who die from TB never receive treatment [2,3,4]. Identification of children at risk of TB due to recent *Mycobacterium tuberculosis* (Mtb) infection, and improvements in diagnostic approaches for pediatric TB, are global public health priorities [1,4].

Progression to TB disease following Mtb infection can be prevented with TB preventive therapy (TPT) [5,6]. Therefore, prompt identification of young, Mtb-infected children, remains the most effective tool to curb TB-related pediatric morbidity and mortality. Currently, there is no reference standard test for asymptomatic Mtb infection. Rather, evidence of immunologic memory to Mtb (referred to as Mtb sensitization) is considered a surrogate for Mtb infection and is used to identify individuals who would most benefit from TPT. Similarly, evidence of Mtb sensitization is often utilized to support a clinical diagnosis of TB in young children undergoing evaluation for TB disease, as microbiologic confirmation of TB remains extremely challenging in this age group [2]. Both the tuberculin skin test (TST) and interferon gamma release assays (IGRAs) can be used to identify individuals with evidence of Mtb sensitization [2,7,8,9,10]. Both tests serve as indicators of immunologic memory to mycobacterial antigens, and neither assay can reliably distinguish between individuals with asymptomatic or subclinical infection and those with active TB disease. A primary limitation of TST is the potential for nonspecific responses following immunization with bacilli Calmette–Guérin (BCG) vaccine and/or exposure to environmental mycobacteria [11,12,13]. Thus, thresholds to determine a positive TST are controversial and dependent on factors such as age, HIV status, nutritional status, country of origin, and TB exposure history [9].

IGRAs, such as the QuantiFERON Gold Plus (QFT-Plus), do not cross-react with antigens present in BCG vaccine or environmental mycobacteria and are more specific than TST for Mtb sensitization [7,8]. Unlike the original QuantiFERON Gold test, the QFT-Plus includes a second antigen tube that captures CD8+ T-cell responses intended to increase the assay’s sensitivity [14], although recent studies have not supported this conclusion [14,15]. The threshold to define a positive IGRA does not consider age, immunocompromising conditions, or TB exposure. For young children, particularly those under 2 years old who are most at risk for rapid progression to TB following exposure, the optimal test to identify those with Mtb sensitization remains controversial [9,14,16], and discordance between TST and IGRA results have been widely reported [17,18]. Identification of young children with evidence of Mtb sensitization is critical to identify asymptomatic children who would most benefit from TPT [19]; it is also critical in clinical decision making for diagnosing TB in unwell children with known TB exposure [20].

Here, we take advantage of a HHC study design where all young children have been recently and heavily exposed to TB in their homes and BCG vaccination is common, to understand how to best identify children with Mtb sensitization using two diagnostic approaches, the TST and QFT-Plus. The objectives of this study were to determine the following: (1) the prevalence of Mtb immune sensitization in young children heavily exposed to TB using both the TST and QFT-Plus and to examine the concordance of these two tests; (2) the diagnostic accuracy of TST and QFT-plus for confirmed and unconfirmed TB in young children. We hypothesized that in a population of young children with extensive exposure to TB, TST and IGRA testing would perform similarly to identify individuals with evidence for Mtb sensitization and would have similar diagnostic accuracy for pediatric TB.

## 2. Materials and Methods

### 2.1. Setting

This study was conducted in Kampala, Uganda, a highly TB-endemic setting [21].

### 2.2. Study Population, Diagnostic Evaluation, and Cohort Assignment

The study population included 130 children of ages <5 years who are household contacts (HHCs) of an adult or adolescent with confirmed pulmonary TB disease (index case). A HHC is defined as a child living in the same home of the index case for ≥7 consecutive days in the past three months. All index cases had symptomatic pulmonary TB disease, confirmed by positive sputum AFB culture (detected on either liquid or solid media) or MTB/RIF GeneXpert^®^ Ultra assay. The parent(s) or guardian(s) of HHCs <5 years were contacted to undergo informed consent within 28 days of the index case’s TB diagnosis. Pediatric HHC enrollment inclusion criteria were as follows: age <60 months old; living in same household as TB index case for at least one week during the three-month period immediately preceding the diagnosis of TB in the index case; resident of the Kampala metropolitan area; and parent/guardian provided written informed consent. Exclusion criteria were as follows: a prior history of being treated for TB; currently receiving treatment for TB; or family was expected to be unavailable for the 12-month follow-up period. The study protocol was approved by the Makerere University School of Biomedical Sciences Research Ethics Committee (SBS-2021-44), the Uganda National Council on Science and Technology (HS1870ES), and the institutional review board at University Hospitals Cleveland Medical Center (STUDY20210373). Written informed consent was obtained from the parent or guardian of each HHC.

After enrollment, each participant underwent a review of medical history and TB symptoms, complete physical examination (including assessment for disseminated and/or extrapulmonary TB), nutritional assessment via anthropometry (height/length, weight, BMI, and upper arm circumference), 2-view chest radiographs (CXR; interpreted by radiologist; Appendix A), and collection of two induced sputum samples for AFB smear, culture, and Xpert Ultra testing, as well as an individual risk assessment. This risk assessment was administered to compute an epidemiologic risk score (ERS) that quantified degree of Mtb exposure. This score ranges from 0 to 10 with a higher score conferring an increased risk. This standard questionnaire included information on risk factors such as degree of contact with the index case, relationship to the index case, index symptoms, and index case clinical characteristics [22]. To estimate the duration of household TB exposure among enrolled children, the number of days of TB-related symptoms among each index case was also collected (Appendix A). At study enrollment, each child underwent HIV testing. Rapid HIV diagnostic testing was performed in all children. HIV-PCR-based confirmatory testing was performed in children <18 months old, recently breastfed infants, and to confirm positive rapid HIV results. Maternal HIV status was established by direct maternal testing if available, or through review of recent test results. Maternal HIV status during pregnancy and/or breastfeeding was also established when possible. Based on maternal and child HIV test results, each child was characterized as HIV-unexposed/uninfected (HUU), HIV-exposed/uninfected (HEU), or a child living-with-HIV (CLWH). BCG vaccination status was determined through documentation of BCG scar on exam and/or provision of BCG vaccination card. QFT-Plus testing and TST placement (interpreted within 2–3 days) were also performed at study enrollment. QFT-Plus was performed in a CAP-certified laboratory and results were interpreted according to the manufacturer’s recommendations [23]. Absolute values for all four tubes (nil; mitogen; TB.1; TB.2) included in the QFT-Plus were also collected. The TST was performed by the Mantoux method (5 tuberculin units per 0.1 mL of purified protein derivative, Tubersol; Connaught Laboratories Limited, Willowdale, ON, Canada). We considered TST positivity using two different cutoffs. First, following the World Health Organization (WHO) guidelines [24], 10 mm of induration indicated a positive TST, except for CLWH or children with severe wasting (defined by length/height-for-weight Z-score ≤−3) when a 5 mm cutoff was used. We derived anthropometric z-scores from WHO child growth standards [25]. Second, we applied a cutoff of 5 mm or greater of induration for all participants, regardless of HIV or nutritional status, as per Center for Disease Control (CDC) guidelines for pediatric TB HHCs [9].

Based on the results of their baseline evaluation, each HHC was assigned to one of two initial cohorts by a study physician using international criteria supported by the WHO for diagnosis of pediatric TB [16,26] and to determine if each child required treatment for TB disease or TPT. HHCs without signs or symptoms of TB disease, and a negative diagnostic evaluation for TB disease, regardless of TST and QFT-Plus results, were considered an asymptomatic exposure cohort. HHCs in the asymptomatic exposure cohort were offered TPT using 6 months of INH treatment, as per WHO recommendations for all pediatric TB HHC <5 yo [26]. HHCs with at least 1 sign or symptom of TB disease and/or CXR findings consistent with pulmonary TB, and/or a positive microbiologic evaluation for TB disease, were considered to have confirmed or unconfirmed TB disease. Children with TB disease received 6 months of TB treatment according to Ugandan national guidelines. Following completion of the 12-month study, children received a final research cohort assignment of asymptomatic Mtb exposure (PedAS) or TB disease (confirmed and unconfirmed; PedTB) using a standardized consensus review process and review of microbiologic data collected at enrollment, and the clinical data collected at enrollment and during the 12-month study period, including scheduled follow-up visits at 1, 3, 6, and 12 months (Appendix A). Children with a positive induced sputum assay for Mtb (by either AFB cultures or Xpert Ultra) were considered to have confirmed TB disease, whereas children meeting research criteria for TB with negative sputum studies were considered to have unconfirmed TB [20]. Children in the PedAS cohort comprised those at enrollment who were negative for the following: TB signs and symptoms, CXR findings consistent with TB disease, and AFB culture and Xpert Ultra sputum test results. Children in the PedAS cohort were not treated for TB disease and did not develop signs or symptoms concerning for TB during the 12-month study period. Thus, TB-specific diagnostics were performed at enrollment only, as no children in the asymptomatic Mtb exposure cohort required repeated investigations for TB during the 12-month period of participation. The overall study design is shown in Figure 1.

### 2.3. Statistical Analyses

We examined the data obtained during the initial TB diagnostic evaluation of pediatric HHCs in order to determine the following: (1) the prevalence of Mtb immune sensitization in young children heavily exposed to TB using both the TST and QFT-Plus, and to examine the concordance of these two tests; and (2) the diagnostic accuracy of TST and QFT-plus for the diagnosis of confirmed and unconfirmed TB (PedTB) in young children at the time of their initial TB diagnostic evaluation. Analysis included the first 130 pediatric HHC to complete the 12-month study. The prevalence of Mtb immune sensitization was determined using TST at both 5 mm and 10 mm thresholds for positivity; manufacturer’s thresholds were utilized to establish QFT-Plus positive, negative, and indeterminant results. This analysis was performed among all participants (full cohort), by age (<2 years and 2–5 years old) and by TB classification (PedAS and PedTB). Correlation analysis between TST and QFT-Plus results was performed, with Cohen’s Kappa coefficient and McNemar’s chi-square test calculations reported. Median results between QFT-Plus tube 1 (TB.1) and tube 2 (TB.2) were compared, and Spearman’s correlation analysis performed. The sensitivity and specificity of TST (at both thresholds) and QFT-Plus for PedTB (using children with confirmed and unconfirmed TB as reference for true TB disease) were calculated. We also conducted univariate analysis to compare final TB classification cohorts (PedAS and PedTB) using demographic information, ERS, HIV classification, BCG status, quantitative and categorical QFT-Plus results, quantitative and categorical (using both 5 mm and 10 mm cutoffs) TST results, and anthropometric z-scores. Comparisons were performed using Fisher’s exact test and chi-square test for categorical variables, and Student’s *t*-test and Mann–Whitney *U* test for continuous variables. Significance was assessed using a 0.05 alpha cutoff. We used the results of this univariate analysis to create a logistic regression model to estimate the odds of being diagnosed with TB. All analyses were performed using R software [27]. Data collection and management for this paper was performed using OpenClinica open-source software (version 3.16. Copyright OpenClinica LLC and collaborators, Waltham, MA, USA, www.OpenClinica.com).

## 3. Results

### 3.1. Demographic and Clinical Characteristics, Diagnostic Results, and TB Classification Among Study Participants

Children were heavily exposed to TB in their homes, with a mean ERS of 7.0 among the entire population (Table 1). The presence of prolonged TB symptoms (cough, fever, sputum production, night sweat, hemoptysis, and weight loss) was common among index cases, with a median duration of 60 days for cough and sputum production (Appendix A). Among all children, 31.5% were QFT-Plus positive, 39.2% were TST positive using a 10 mm threshold, and 49.2% were TST positive using a 5 mm threshold (Table 1). Among 130 HHCs, we classified 75 as PedAS and 55 as PedTB. The majority of 55 HHCs classified as PedTB (90.9%) had findings consistent with TB on CXR (Table 2). Notably, all five children with normal CXR findings in the PedTB cohort had microbiologically confirmed TB. One-third of children with a final classification of PedTB presented with symptoms consistent with TB (Table 2), and five had confirmed TB (Table 2). During their baseline evaluation, 13 children who did not received a final classification of PedTB were noted to have abnormal CXR and/or symptoms such as cough and fever. Among these 13 children, 9 were treated with an empiric course of antibiotics for non-TB pneumonia at their initial study visit with resolution of symptoms, and 4 children were provided symptomatic care (e.g., anti-pyrectics, anti-histamines, steroids, zinc) for suspected viral and/or asthma-related illnesses. All 13 children had resolution of their initial illness observed during follow-up and no further signs or symptoms consistent with pediatric TB emerged during the subsequent 12-month follow-up period. In summary, among 130 heavily exposed child household contacts, less than half tested positive on either the TST or QFT-Plus.

### 3.2. Concordance Between TST and QFT-Plus Among Study Participants

Concordance analysis was performed, comparing results of TST (at both 10 mm and 5 mm thresholds) and QFT-Plus among all children at study enrollment. Five children (3.7%) had indeterminate QFT-Plus results and were excluded from subsequent analysis; indeterminate results were due to failure of mitogen to elicit a response (N = 3) or a high background response (N = 2). Using the recommended threshold for interpretation, QFT-Plus positivity rate was 31.5% across the entire cohort. At the 10 mm threshold, TST and QFT-Plus exhibited substantial agreement (Cohen’s Kappa = 0.74), which was higher in children 2–5 versus children <2 years old (0.83 versus 0.60, respectively). At the 5 mm threshold, TST and QFT-Plus exhibited moderate agreement (Cohen’s Kappa = 0.59), which was higher in children 2–5 versus children <2 years old (0.72 versus 0.42, respectively). Figure 2 visualizes agreement between the two testing approaches among all children, and Table 3 and Table 4 detail the concordance between TST and QFT-Plus results in all children using a 10 mm and 5 mm threshold for positivity, respectively. Agreement between the two tests was highest among children in the PedAS cohort, regardless of age (Appendix A). Overall, TST and QFT-Plus testing showed acceptable agreement, particularly in the PedAS cohort and children 2–5 years old.

### 3.3. Diagnostic Accuracy of TST and QFT-Plus for Pediatric TB

We first compared demographic and clinical characteristics between children in the PedAS and PedTB cohorts. Here, we found no statistically significant differences in age, sex, ERS, HIV exposure or infection, BCG status or anthropometric measurements between cohorts (Table 1 and Appendix A).

We then calculated the sensitivity and specificity of TST and QFT for TB disease (confirmed and unconfirmed), as shown in Table 5. A TST cutoff at 5 mm detected 60% of children with confirmed and unconfirmed TB. Raising the TST threshold to 10 mm decreased sensitivity, allowing more than half of TB cases to be missed. QFT-Plus testing yielded the highest specificity (0.72) but lowest sensitivity (0.36) for TB. There were no significant differences in categorical or quantitative QFT-Plus results between PedAS and PedTB cohorts, including when quantitative values were compared only among those children with a positive qualitative QFT-Plus result (Table 1). Notably, the PedTB cohort had a higher quantitative TST result than the PedAS cohort (8.6 mm vs. 0 mm, respectively, *p* = 0.02) and a greater rate of TST positivity with the 5 mm cutoff when compared to the PedAS cohort (61.1% vs. 39.7%, respectively, *p* = 0.02). When using the 10 mm cutoff, the TST positivity rate was still greater in the PedTB cohort versus the PedAS cohort (47.3% vs. 33%, respectively, *p* = 0.15), but this difference did not reach statistical significance (Table 1).

### 3.4. Relationship Between Age and Assessments for Mtb Sensitization Among Children with and Without TB

Next, we explored the relationship between age and the positivity rate for the TST and QFT-Plus (Table 6). Among children ages 2–5 years, 50% of those classified as TB (PedTB) were QFT-Plus positive, and TST positivity was 56.3% versus 67.7% (10 mm versus 5 mm thresholds, respectively). Conversely, QFT-Plus and TST positivity rates were below 36% in those without TB (PedAS). Among children <2 years, our analysis was limited as only 20% of the children diagnosed with TB demonstrated a positive QFT-Plus (Table 3). The TST positivity rate among these younger children with TB was 34.8% when using a 10 mm threshold and 52.2% when using a 5 mm threshold. Therefore, TST consistently yielded higher positivity rates than the QFT-Plus among PedTB children in both age groups.

### 3.5. Role of TST in Predicting TB

Next, we assessed the value of tests of Mtb immune sensitization in predicting TB disease. Based on univariate analysis (Table 1), both the 5 mm TST categorical result and quantitative TST result could be included as predictors in our logistic regression model, with equivalent model fit statistics. Due to collinearity, we decided to include the categorical TST variable only while excluding the quantitative TST variable and the categorical QFT-Plus variable. Finally, we adjusted the model for age (months), sex, HIV, and BCG status (Appendix A).

As shown in Table 7, the odds of TB disease were approximately two times higher when the child had a positive TST using the 5 mm cutoff (*p* = 0.04). Using the same model, we re-evaluated the odds of TB disease when applying the 10 mm cutoff. Here, the odds of TB disease were not significantly different based on TST result. Thus, only the 5 mm TST threshold—unlike the 10 mm cutoff—showed significant predictive value for TB disease.

### 3.6. Contribution of QFT-Plus Tube 1 Versus Tube 2 in Identification of Children with Mtb Immune Sensitization

We examined the role of QFT-Plus Tube TB.1 and Tube TB.2 in identification of children with Mtb immune sensitization. Among all participants, there was not a significant difference in median production of IFN-gamma between Tube TB.1 [median 0.01 IU/mL (IQR, 0–1.85)] and Tube TB.2 [median 0.02 IU/mL (IQR, 0–3.010–3.01; Appendix A). Furthermore, we demonstrated significant correlation between Tube TB.1 and Tube TB.2 (R = 0.82; *p* < 0.001). Only three children had discordant QFT-Plus Tube TB.1 and Tube TB.2 results in Figure 3 (Appendix A). Similarly, among the 41 participants with a positive QFT-Plus result, no significant differences in the median production of IFN-gamma between Tube TB.1 [median 6.12 IU/mL (IQR, 2.68–9.01)] and Tube TB.2 [median 6.14 IU/mL (IQR, 3.82–9.92)] were found. The correlation between Tube TB.1 and Tube TB.2 (R = 0.86; *p* < 0.001) also remained strong. The near identical IFN-gamma responses and the strong correlation between Tube TB.1 and Tube TB.2 suggests the addition of a second tube adds minimal incremental value for detecting Mtb sensitization in young children.

## 4. Discussion

Identification of young children with evidence for Mtb immune sensitization is critical to identify those who would most benefit from TPT and to support a clinical diagnosis of TB among TB-exposed children with concerning signs and symptoms. Here, we employed a TB HHC study that included all children under 5 years living with an individual with microbiologically confirmed pulmonary TB to compare the capacity of TST and QFT-Plus testing to identify children with evidence of Mtb sensitization. We found that despite the high ERS observed in this population of children, and prolonged cough and sputum production among index cases indicating extensive household TB exposure, TST and QFT-Plus testing demonstrated evidence of Mtb sensitization in less than 50% of all exposed children. Overall, a 5 mm TST threshold identified the most children with evidence of Mtb sensitization, and the enhanced performance of the TST was most pronounced in young children with TB. The use of a second Mtb-specific test condition (TB.2) in the commercial QFT-Plus assay did not significantly increase the detection of Mtb immune sensitization in this cohort of young children.

Young children less than 5 years old are uniquely vulnerable to develop TB following recent Mtb infection, with progression to TB within 24 months of exposure in up to 20% of children [19,28,29]. A young age is also associated with more severe clinical manifestations of TB and substantially higher mortality rates [29,30]. Therefore, the identification of young children at risk of TB disease due to recent Mtb infection, and improvement of diagnostic approaches for pediatric TB, remain global public health priorities [1,4]. Unfortunately, our findings suggest that neither the TST nor QFT-Plus are reliable indicators of Mtb sensitization in a population of young Ugandan children with extensive TB exposure, particularly those under 2 years old. Our findings are consistent with previous studies where both tests had low sensitivity in young children with TB [31,32,33,34,35] or suspected disease [34,35,36]. Although the sample size is limited here, our findings highlight potential limitations of immune-based diagnostics, such as IGRAs, in young children being evaluated for Mtb infection and TB disease. Although the introduction of a second tube containing Mtb-specific peptides (TB.2) in the QFT-Plus was anticipated to enhance detection of immune sensitization to Mtb, we found results to be highly correlated between TB.1 and TB.2, as reported in other studies in children [14,15]. Given that additional blood volumes are required to perform both TB.1 and TB.2 conditions, the utility of this testing approach in young children should be reconsidered.

In the United States, where BCG vaccination is not routinely given, the CDC and American Academy of Pediatrics recommends a threshold of 5 mm for children with known TB exposure if TST is performed [9,37]. However, IGRAs have now become the preferred test for detection of Mtb sensitization in well-resourced settings, particularly among individuals 5 years and older [9]. In these populations, the enhanced specificity of IGRAs compared to TST can reduce unnecessary exposure to TPT among TST-positive BCG-vaccinated individuals [9,38]. The potential for false positive results among young children with recent BCG vaccinations was highlighted in a large recent observational study of high-risk individuals living in the US [39]. In addition, several studies in young children suggest that BCG vaccination is associated with a positive TST among individuals without risk factors for Mtb exposure, and that this association wanes with increasing age [40,41,42,43].

Concerns about false-positive TST results among BCG-vaccinated young children is a driving factor behind the WHO’s recommendation to apply a 10 mm threshold for positivity among children-without-HIV-infection and non-wasted children known to be exposed to TB [16]. One large study performed in South Africa demonstrated that IGRA results are more highly correlated with measures of TB exposure using the ERS as compared to TST positivity [44]. Moreover, among South African children of 2–5 years, the quantitative IFN-gamma response was strongly associated with TB disease status [45]. In this current cohort of young Ugandan children, however, we did not observe either of these findings. Currently in Uganda, WHO guidelines using a 10 mm threshold for positive TST interpretation are followed. However, our findings suggest that application of a 5 mm threshold for all TB HHCs under 5 years would substantially increase the identification of children with evidence for Mtb sensitization, particularly those being evaluated for TB disease. Among asymptomatic, TB-exposed young children, a TST threshold of 5 mm could be utilized to prioritize children for TPT when universal treatment is not feasible (despite WHO recommendations for universal TPT for TB-exposed HHC under 5 years) [26].

As per international recommendations, negative IGRA-based testing cannot be considered a rule out test for pediatric TB [14,18]. Our study re-enforces this recommendation, with QFT-Plus testing having extremely limited sensitivity and sub-optimal specificity for pediatric TB. False-negative IGRA-based testing could be driven by underlying host, technical, and/or environmental influences [46]. Discordance between positive TST and negative IGRA-based evaluations for Mtb sensitization could be driven by underlying immunologic responses unique to pediatric Mtb infection and TB [47]. We hypothesize that immune sensitization to Mtb infection in young children is not sufficiently identified using quantification of interferon gamma production alone in response to Mtb-specific peptides. Several studies among asymptomatic Mtb-exposed children, and children being evaluated for TB disease, support that quantifying production of non-IFN-gamma, alternative cytokines, in response to Mtb-specific antigens, is a promising approach to identify children with Mtb sensitization and potentially delineate between infection and disease states [48,49,50]. Discordance with negative TST and positive QFT-Plus-based testing in this study was rare and observed in two children with asymptomatic exposure and two children with TB. False negative TST results have often been ascribed to technical failures in placement and/or interpretation, or anergy among individuals with symptomatic disease [51].

Our study has several limitations. Firstly, our overall sample size and statistical power were limited, particularly for younger children with confirmed TB. We also did not have access to a sufficient number of BCG-unvaccinated, TB HHCs under 5 years old to assess associations between BCG vaccination and TST results using 5 mm and 10 mm thresholds. Similarly, the number of CLWH was limited and comparisons between children living with- and without-HIV could not be performed. We did not have access to a similarly aged population of non-TB exposed Ugandan children to assess the specificity of TST versus QFT-Plus for Mtb sensitization. As is expected among children treated for TB disease, the majority of children did not demonstrate microbiologically confirmed disease, despite testing two induced sputum samples for Mtb using AFB smear, AFB liquid and solid cultures, and Xpert Ultra [52,53,54]. Sputum induction was selected as it has been shown to be safe, practical, and have similar yields for disease confirmation in young children as compared to first-morning gastric aspirates [55,56,57]. There remains no gold standard for asymptomatic Mtb infection and/or exposure, limiting all assessments of the diagnostic accuracy of the TST and QFT-Plus for these indications. We note that in our study design, TST results were available in real-time, whereas QFT-Plus results were delayed, and this could have biased an initial assignment of TST-positive children to the TB disease cohort. However, our final cohort assignments (completed following a 12-month period of observation) demonstrated that all children classified as having TB disease had either an abnormal CXR suggestive of pulmonary TB, microbiologic-confirmed disease, and/or signs and symptoms of TB disease; no child was classified as having TB disease solely upon the basis of their exposure history and a positive TST. Finally, TST and QFT-Plus testing was performed at study entry and not repeated, and we cannot rule out that some children may have been in the window period prior to development of Mtb-specific adaptive immune responses. However, given that children were living in homes where index cases reported 1–2 months of TB symptoms, it is unlikely that repeat testing would have identified additional children with evidence of immune sensitization.

## 5. Conclusions

Overall, in a population of young Ugandan children with extensive, recent, household exposure to TB, the TST identified more children with evidence for Mtb immune sensitization and TB disease than QFT-Plus testing. The findings from this study are highly relevant for young children who are TB HHCs in an endemic setting with high rates of BCG vaccination, limited resources to evaluate for Mtb infection and TB disease, and poor access to TPT. In resource limited settings where TB is endemic and access to healthcare limited, children under 5 years are extremely vulnerable to severe TB disease if not identified promptly, and remain the most challenging population to diagnose. In these settings, we recommend that public health programs endorse TST testing to assess for Mtb sensitization in this age group, and that a 5 mm threshold be applied to asymptomatic children with known Mtb exposure for prioritization of TPT and among Mtb-exposed children being evaluated for TB disease.

## Figures and Tables

**Figure 1 pathogens-14-00924-f001:**
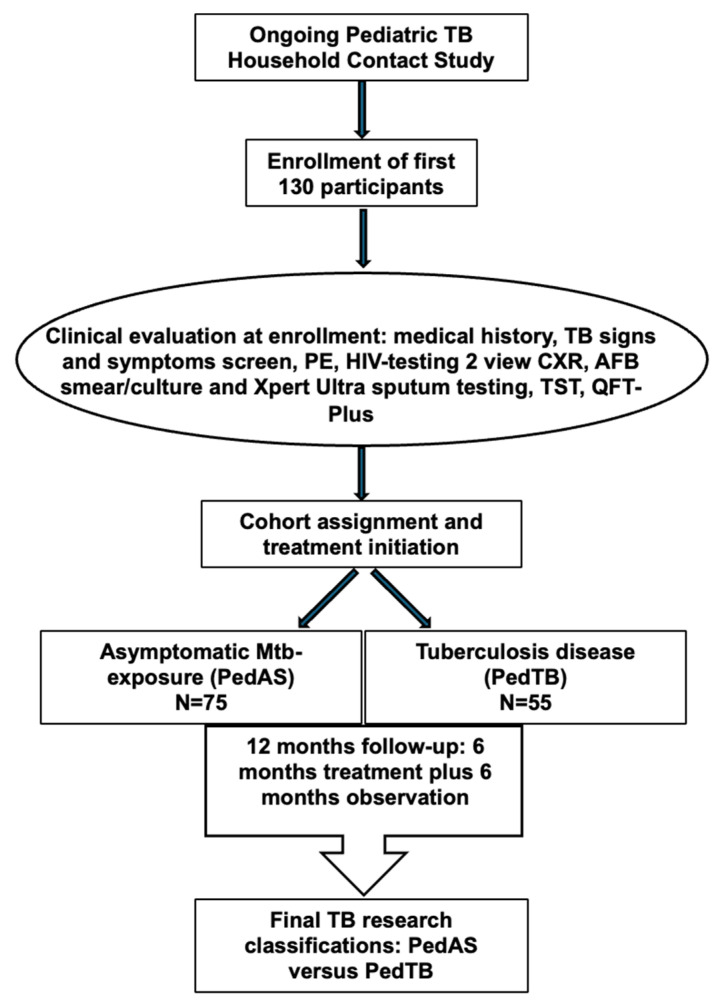
Overview of pediatric TB household contact study.

**Figure 2 pathogens-14-00924-f002:**
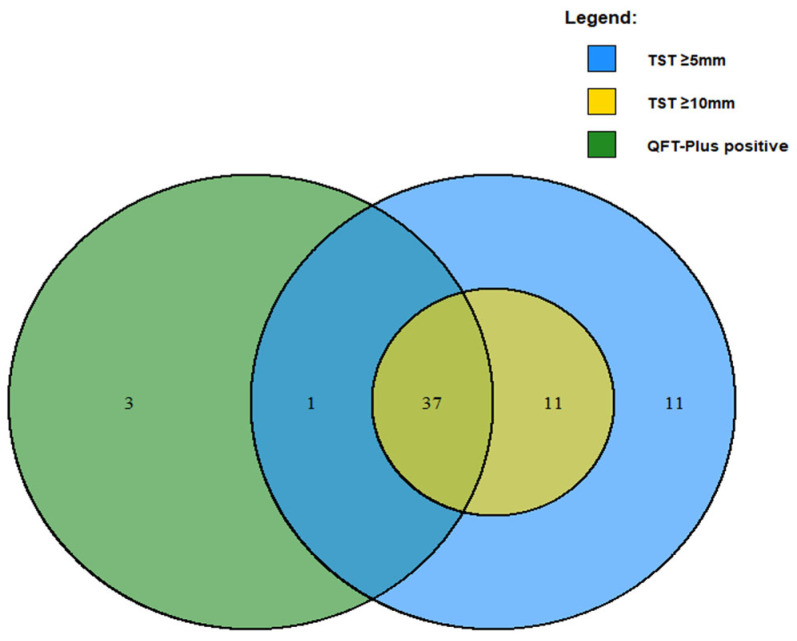
Venn diagram illustrates concordance between TST testing and QFT-Plus testing at study enrollment. All children were evaluated for Mtb immune sensitization at enrollment using both TST and QFT-Plus testing. Areas of overlap illustrate concordance between the tests, using either a 5 mm (blue) or 10 mm (yellow) thresholds for TST positivity.

**Figure 3 pathogens-14-00924-f003:**
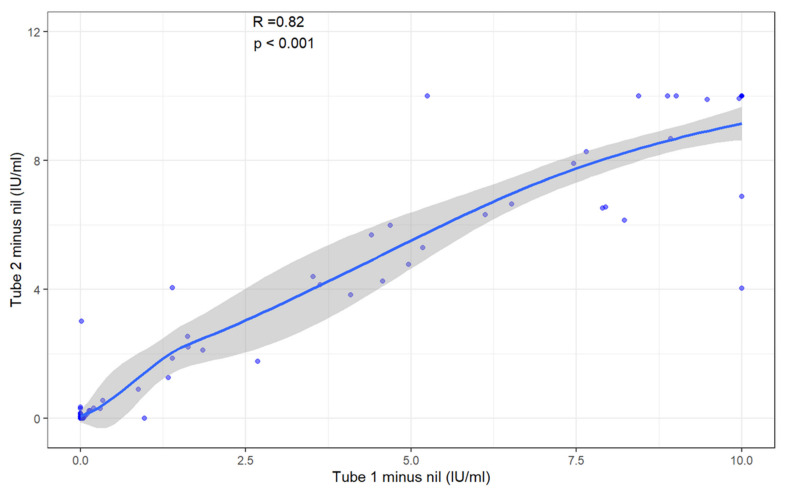
Correlation between quantitative QFT-Plus Tube TB.1 and Tube TB.2 results. Using Spearman’s correlation coefficient (rho), we compared the results of TB.1 and TB.2. The results of TB.1 and TB.2 were strongly and positively correlated to each other based on a rho of 0.82 (*p* ≤ 0.001). Analysis did not include five participants with indeterminate results.

**Table 1 pathogens-14-00924-t001:** Demographic characteristics, QFT-Plus, and TST results of study participants with and without TB.

	Full Cohort	PedAS	PedTB	*p*-Value (Test)
*n*	130	75	55	
Age (months)	27.5 [±6.2]	25.9 [±16.2]	29.6 [±16.2]	0.20 (*t*-test)
Sex = male	73 (56.2)	42 (56.0)	31 (56.4)	1.00 (χ^2^)
Epidemiologic Risk Score (range: 0–10)	7 [6.0–8.0]	7 [6.0–9.0]	7 [6.0–8.0]	0.12 (MWU)
HIV classification				0.86 (Fisher’s)
HIV unexposed uninfected (HUU)	87 (66.9)	48 (64.0)	39 (70.9)	
HIV exposed uninfected (HEU)	19 (14.6)	12 (16.0)	7 (12.7)	
Child living with HIV	4 (3.1)	2 (2.7)	2 (3.7)	
HIV exposure unknown HIV result pending ^a^	19 (14.6) 1 (0.8)	12 (16.0) 1 (1.3)	7 (12.7) 0 (0)	
IGRA (QFT-Plus) = positive ^b^	41 (31.5)	21 (28.0)	20 (36.4)	0.34 (χ^2^)
Mean quantitative IGRA result ^c^	0.01 [0–2.47]	0.02 [0–1.77]	0.03 [0–4.96]	0.75 (MWU)
Mean quantitative IGRA result ^c^ in children with positive IGRA (*n* = 41)	6.59 [2.72–9.22]	5.34 [1.98–9.51]	6.89 [4.63–7.97]	0.82 (MWU)
TST = positive (10 mm)	51 (39.2)	25 (33.3)	26 (47.3)	0.15 (χ^2^)
TST = positive (5 mm)	64 (49.2)	31 (39.7)	33 (61.1)	0.02 * (χ^2^)
Quantitative TST result	3.4 [0–14.9]	0 [0–13.5]	8.6 [0–16.3]	0.02 * (MWU)
BCG scar present	112 (86.2)	64 (85.3)	48 (87.3)	0.80 (Fisher’s)

PedAS: asymptomatic Mtb exposure; PedTB: TB disease; Counts (percentages), means [± standard deviation], or median [quartiles]; χ^2^: chi-squared test; MWU: Mann–Whitney U test; QFT-Plus: QuantiFERON-TB Gold Plus; ᵃ Child will undergo testing after breastfeeding is complete; ^b^ Does not include 5 participants with indeterminate results; ^c^ Calculated as (QFT-Plus tube 1 + QFT-Plus tube 2)/2; * Statistically significant at *p* < 0.05.

**Table 2 pathogens-14-00924-t002:** Summary of diagnostic evaluation for TB among study participants.

	PedAS	PedTB
*n*	75	55
Chest X-ray (CXR)		
Normal	70 (93.3%)	5 (9.1%)
Moderately advanced disease ^a^ Unknown	3 (4.0%) 2 (2.7%)	50 (90.9%) 0 (0%)
TB symptoms		
No symptoms	62 (82.7%)	36 (65.5%)
At least one symptom	13 (17.3%)	19 (34.5%)
Extra-pulmonary TB symptoms		
No symptoms	75 (100%)	55 (100%)
At least one symptom	0 (0%)	0 (0%)
Xpert Ultra result		
Negative	75 (100%)	51 (92.7%)
Positive Unknown	0 (0%) 0 (0%)	3 (5.5%) 1 (1.8%)
Smear Microscopy result		
Negative	74 (98.7%)	53 (96.4%)
Positive Unknown	0 (0%) 1 (1.3%)	2 (3.6%) 0 (0%)
Culture result		
Negative	75 (100%)	50 (90.9%)
Positive	0 (0%)	5 (9.1%)

PedAS: asymptomatic Mtb exposure; PedTB: TB disease; ^a^ moderately advanced disease—disease may be present in one or both lungs; the total extent must not be more than the following: (a) scattered lesions of slight to moderate density may not involve more than total volume of one lung or the equivalent volume of both lungs. (b) dense, confluent lesions may not involve more than 1/3 of the volume of one lung. (c) the total diameter of cavity(ies) may not be greater than 4 cm.

**Table 3 pathogens-14-00924-t003:** Cross-tabulation of QFT-Plus and TST results (10 mm) for all participants.

		TST Results (10 mm)			
**Full Cohort**	**QFT-Plus Results**	**N = 125 ***	**Positive**	**Negative**	
		**Positive**	37 (29.6%)	4 (3.2%)	McNemar χ^2^ = 0.12
		**Negative**	11 (8.8%)	73 (58.4%)	Cohen’s Kappa ^a^ = 0.74
**Children <2 years of age**	**QFT-Plus Results**	**N = 53**	**Positive**	**Negative**	
		**Positive**	11 (20.7%)	1 (1.9%)	McNemar χ^2^ = 0.04
		**Negative**	8 (15.1%)	33 (62.3%)	Cohen’s Kappa ^a^ = 0.60
**Children 2–5 years** **of age**	**QFT-Plus Results**	**N = 72**	**Positive**	**Negative**	
		**Positive**	26 (36.0%)	3 (4.2%)	McNemar χ^2^ = 1.0
		**Negative**	3 (4.2%)	40 (55.6%)	Cohen’s Kappa ^a^ = 0.83

* Does not include 5 participants with indeterminate QFT results; ^a^ Cohen’s Kappa interpretation: 0.01–0.20 as none to slight, 0.21–0.40 as fair, 0.41–0.60 as moderate, 0.61–0.80 as substantial, and 0.81–1.00 as almost perfect agreement.

**Table 4 pathogens-14-00924-t004:** Cross-tabulation of QFT and TST results (5 mm) for all participants.

		TST Results (5 mm)			
**Full Cohort**	**QFT-Plus Results**	**N = 125 ***	**Positive**	**Negative**	
		**Positive**	38 (30.4%)	3 (2.4%)	McNemar χ^2^ = 0.0003
		**Negative**	22 (17.6%)	62 (49.6%)	Cohen’s Kappa ^a^ = 0.59
**Children <2 years of age**	**QFT-Plus Results**	**N = 53**	**Positive**	**Negative**	
		**Positive**	11 (20.8%)	1 (1.9%)	McNemar χ^2^ = 0.002
		**Negative**	14 (26.4%)	27 (50.9%)	Cohen’s Kappa ^a^ = 0.42
**Children 2–5 years** **of age**	**QFT-Plus Results**	**N = 72**	**Positive**	**Negative**	
		**Positive**	27 (37.5%)	2 (2.8%)	McNemar χ^2^ = 0.11
		**Negative**	8 (11.1%)	35 (48.6%)	Cohen’s Kappa ^a^ = 0.72

* Does not include 5 participants with indeterminate QFT-Plus results; ^a^ Cohen’s Kappa interpretation: 0.01–0.20 as none to slight, 0.21–0.40 as fair, 0.41–0.60 as moderate, 0.61–0.80 as substantial, and 0.81–1.00 as almost perfect agreement.

**Table 5 pathogens-14-00924-t005:** Sensitivity and specificity of tuberculin skin test (TST) and QuantiFERON-Plus (QFT-Plus) test for TB disease.

	Tuberculosis (*n* = 55)	Asymptomatic Exposure (*n* = 75)	Sensitivity	Specificity
TST positive (5 mm)	33	31	0.6	0.59
TST negative (5 mm)	22	44		
TST positive (10 mm)	26	25	0.47	0.67
TST negative (10 mm)	29	50		
QFT-Plus positive	20	21	0.36	0.72
QFT-Plus negative	35	54		

**Table 6 pathogens-14-00924-t006:** Assessment of Mtb immune sensitization among children with and without TB stratified by age.

		Under 2 Years of Age (*n* = 57)			2 to 5 Years of Age (*n* = 73)		
		PedAS	PedTB		PedAS	PedTB	
IGRA ^a^ (QFT-Plus)	Positive	8 (24.2%)	4 (20%)	*p* = 0.98	13 (35.6%)	16 (50%)	*p* = 0.21
	Negative	25 (75.8%)	16 (80%)		27 (67.5%)	16 (50%)	
TST (10 mm)	Positive	13 (38.2%)	8 (34.8%)	*p* = 1.00	12 (29.3%)	18 (56.3%)	*p* = 0.04 *
	Negative	21 (61.8%)	15 (65.2%)		29 (70.7%)	14 (43.8%)	
TST (5 mm)	Positive	16 (47.1%)	12 (52.2%)	*p* = 0.91	15 (35.7%)	21 (67.7%)	*p* = 0.01 *
	Negative	18 (52.9%)	11 (47.8%)		27 (64.3%)	10 (32.3%)	

PedAS: asymptomatic Mtb exposure; PedTB: TB disease; QFT-Plus: QuantiFERON-TB Gold Plus; ^a^ does not include 5 children with indeterminate QFT-Plus results; * statistically significant.

**Table 7 pathogens-14-00924-t007:** Logistic regression models demonstrating odds of TB disease using 5 mm and 10 mm TST thresholds.

Covariates	Adjusted OR	95% CI	*p*-Value
TST results (5 mm)			
Negative	(Ref.)	(Ref.)	(Ref.)
Positive	2.09	1.02–4.37	0.04 *
Sex = male	0.85	0.40–1.82	0.68
Age (months)	1.02	0.99–1.04	0.15
HIV = positive	0.80	0.04–9.22	0.86
BCG scar = present	1.19	0.42–3.58	0.75
TST results (10 mm)			
Negative	(Ref.)	(Ref.)	(Ref.)
Positive	1.71	0.83–3.55	0.15
Sex = male	0.87	0.41–1.85	0.73
Age (months)	1.02	0.99–1.04	0.20
HIV = positive	0.72	0.03–8.16	0.79
BCG scar = present	1.29	0.46–3.85	0.63

* Statistically significant at *p* < 0.05; CI: confidence interval.

## Data Availability

The datasets used and/or analyzed during the current study are available from the corresponding author on reasonable request.

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
