# Peer review of "Immune Sensitization to *Mycobacterium tuberculosis* Among Young Children with and without Tuberculosis"

_pathogens, 2025, doi:10.3390/pathogens14090924_

Round 1

Reviewer 1 Report

Comments and Suggestions for Authors

The Introduction section does not provide an understanding of the authors' purpose for conducting this study. Comparisons of the diagnostic value of the tuberculin skin test and QFT-Plus in children under five years of age have been repeatedly conducted and are reflected in review publications, see, for example, DOI: 10.1097/INF.0000000000003877. The Abstract section states that the authors 1) measured the "usefulness" of the tuberculin skin test and QFT-Plus for identifying children with signs of sensitization to Mtb; 2) measured the probability of developing tuberculosis using a logistic regression model. In this section, the authors write about the comparison of the "usefulness" of the tuberculin skin test and QFT-Plus as the purpose of the study. It is necessary that the description of the purpose of the study be identical in both sections.

Materials and Methods section.

A clearer presentation of the study design is required here. The description of the methods does not fully correspond to the obtained results (for example, the results from Figure 2. Correlation between quantitative QFT-Plus Tube TB.1 and Tube TB.2 results). For some reason, the authors placed the description of this stage of the research in the subsection with statistical analysis methods.

In subsection 2.2 (Population), the criteria for including patients in the study should be clearly formulated. Were there any children who received the BCG vaccine? Below, in subsection 2.3, it is said that the study included patients who were tested for HIV infection. It seems that it would be more correct to move the division of patients by HIV infection from subsection 2.3 here, to subsection 2.2. There is also a division of patients by age, but this subsection does not describe it, but it suddenly appears in the subsection describing statistical methods.

Is it possible to structure subsection 2.3 (Design)? For example, make separate descriptions of the research methods (at least separate the methods for detecting sensitization to the tuberculosis bacillus) and the principles for forming subgroups? Or is it possible to make a flow chart describing the design of this study? The reviewer also suggests moving the population/demographic parameters to subsection 2.2.

The authors used a logistic regression model to estimate the probability of developing tuberculosis. The authors describe the diagnostics of sensitization to Mtb and then immediately say about the diagnostics of tuberculosis, based on the results obtained here, isoniazid is prescribed (or not prescribed). Later, after 12 months, the diagnostics of tuberculosis were performed again. For what purpose? Did treatment with isoniazid affect the development of tuberculosis? Which diagnostic results were used to create the model? The reviewer believes that it is necessary to make the description of this stage of the study more detailed.

Results section.

The author's interpretation of the results obtained in this study is certainly necessary. Many tables provide an understanding of the results; however, having a short conclusion at the end of each subsection would significantly improve the reader's comprehension of the text.

Author Response

We sincerely thank the reviewers for their thoughtful and constructive comments. We greatly appreciate the time and effort invested in reviewing our manuscript. Below, we provide detailed responses to each point raised, along with corresponding revisions made to the manuscript.

Reviewer #1

The Introduction section does not provide an understanding of the authors' purpose for conducting this study. Comparisons of the diagnostic value of the tuberculin skin test and QFT-Plus in children under five years of age have been repeatedly conducted and are reflected in review publications, see, for example, DOI: 10.1097/INF.0000000000003877. The Abstract section states that the authors 1) measured the "usefulness" of the tuberculin skin test and QFT-Plus for identifying children with signs of sensitization to Mtb; 2) measured the probability of developing tuberculosis using a logistic regression model. In this section, the authors write about the comparison of the "usefulness" of the tuberculin skin test and QFT-Plus as the purpose of the study. It is necessary that the description of the purpose of the study be identical in both sections.

             We have re-written both the abstract and introduction to clearly state the purpose for conducting our study.  Specifically, this study was conducted to determine: 1) the prevalence of Mtb-immune sensitization in young children heavily exposed to TB using both the tuberculin skin test (TST) and QuantiFERON Gold Plus (QFT-Plus) interferon-gamma release assay, and to examine the concordance of these two tests; and 2) the diagnostic accuracy of TST and QFT-plus for confirmed and unconfirmed TB in young children. [Lines 108-111]

Materials and Methods section.

A clearer presentation of the study design is required here. The description of the methods does not fully correspond to the obtained results (for example, the results from Figure 2. Correlation between quantitative QFT-Plus Tube TB.1 and Tube TB.2 results).

            The methods section has been substantially revised based on the reviewer's suggestions.

For some reason, the authors placed the description of this stage of the research in the subsection with statistical analysis methods.

            The methods section, including description of correlation analysis for TB.1 and TB.1, has been revised to improve clarity.

In subsection 2.2 (Population), the criteria for including patients in the study should be clearly formulated. Were there any children who received the BCG vaccine? Below, in subsection 2.3, it is said that the study included patients who were tested for HIV infection. It seems that it would be more correct to move the division of patients by HIV infection from subsection 2.3 here, to subsection 2.2. There is also a division of patients by age, but this subsection does not describe it, but it suddenly appears in the subsection describing statistical methods.

            We thank the reviewer for this suggestion and have restructured our methods section as suggested, including the section detailing HIV testing.

Is it possible to structure subsection 2.3 (Design)? For example, make separate descriptions of the research methods (at least separate the methods for detecting sensitization to the tuberculosis bacillus) and the principles for forming subgroups? Or is it possible to make a flow chart describing the design of this study? The reviewer also suggests moving the population/demographic parameters to subsection 2.2.

                The methods section has been substantially revised based on the reviewer's suggestions. A flow chart that visualized the overall study design has also been added to the revised manuscript (Figure 1).

The authors used a logistic regression model to estimate the probability of developing tuberculosis. The authors describe the diagnostics of sensitization to Mtb and then immediately say about the diagnostics of tuberculosis, based on the results obtained here, isoniazid is prescribed (or not prescribed). Later, after 12 months, the diagnostics of tuberculosis were performed again. For what purpose? Did treatment with isoniazid affect the development of tuberculosis? Which diagnostic results were used to create the model? The reviewer believes that it is necessary to make the description of this stage of the study more detailed.

            We apologize for the confusion with this section of the manuscript and have revised the methods section to more clearly describe when and how TB diagnostics and classifications were performed. As detailed in the revised methods section:  

"Following completion of the 12-month study, children received a final research cohort assignment of asymptomatic Mtb-exposure (PedAS) or TB disease (confirmed and unconfirmed; PedTB) using a standardized consensus review process and review of microbiologic data collected at enrollment, and the clinical data collected at enrollment and during the 12-month study period, including scheduled follow-up visits at 1, 3, 6, and 12 months (Supp Table 2). Children with a positive induced sputum assay for Mtb (by either AFB cultures or Xpert Ultra) were considered to have confirmed TB disease, whereas children meeting research criteria for TB with negative sputum studies were considered to have unconfirmed TB (19). Children in the PedAS cohort were those at enrolment who were negative for: TB signs and symptoms, CXR findings consistent with TB disease, and AFB culture and GeneXpert sputum test results. Children in the PedAS cohort were not treated for TB disease and remained asymptomatic for TB during the 12-month study period. Thus, TB-specific diagnostics were performed at enrollment only, as no children in the asymptomatic Mtb-exposure cohort required additional investigations for TB during the 12-month period of participation." [Lines 178-192]

Results section.

The author's interpretation of the results obtained in this study is certainly necessary. Many tables provide an understanding of the results; however, having a short conclusion at the end of each subsection would significantly improve the reader's comprehension of the text.

            We thank the reviewer for this suggestion and have revised the results section to include summary statements.

Reviewer 2 Report

Comments and Suggestions for Authors

The paper addresses an important public health issue - detection of Mycobacterium tuberculosis (Mtb) sensitization in children under 5 years old - and compares the diagnostic utility of TST and QFT-Plus in a Ugandan household contact cohort. The topic is relevant, especially given the persistent challenges in diagnosing TB infection in young children. The study’s dataset and context are valuable, but several areas need strengthening for clarity, reproducibility, and impact. A few comments to be addressed:

  • While the introduction describes the importance of diagnosing Mtb in children, the precise hypothesis could be more explicitly stated. Is the aim to compare test sensitivity, agreement, or predictive value for TB disease progression?
  • The paper notes cases where TST and QFT-Plus results differ, but the clinical implications of these discordances are not deeply explored. Discussion could consider biological and operational reasons (e.g., immunosuppression, test variability).
  • The operational definition (positive TST and/or QFT-Plus) should be clearly stated upfront, and any rationale for thresholds (e.g., ≥5mm induration) should be provided.
  • With 130 children, subgroup analyses (especially for TB disease vs. no disease) may be underpowered. A power calculation or discussion of statistical limitations would strengthen credibility.
  • The applicability of findings to other high-burden or low-burden settings could be addressed more directly.

Minor comments:

  • Ensure all tables are self-contained with clear legends, sample sizes, and units.

  • The agreement between TST and QFT-Plus could be visually summarized (e.g., Venn diagram or Bland–Altman plot).

  • Minor typographical errors should be corrected (e.g., “Univeristy” → “University”).

  • Standardize abbreviations and ensure each is defined upon first use.

  • Some key references on pediatric immune response to Mtb and the limitations of IGRAs in young children could be added.

  • WHO guidelines on pediatric TB diagnostics should be cited.As general comments please consider the following: 

  • Include kappa statistics to quantify agreement between TST and QFT-Plus.
  • Provide sensitivity and specificity estimates using TB disease as a reference (even if imperfect).

  • Expand on programmatic implications: Would findings suggest one test over another for Ugandan children? Are there cost or logistical considerations?

  • Consider adding a limitations section explicitly acknowledging small sample size, potential misclassification, and absence of a gold standard for Mtb infection.

Author Response

We sincerely thank the reviewers for their thoughtful and constructive comments. We greatly appreciate the time and effort invested in reviewing our manuscript. Below, we provide detailed responses to each point raised, along with corresponding revisions made to the manuscript.

Reviewer #2

The paper addresses an important public health issue - detection of Mycobacterium tuberculosis(Mtb) sensitization in children under 5 years old - and compares the diagnostic utility of TST and QFT-Plus in a Ugandan household contact cohort. The topic is relevant, especially given the persistent challenges in diagnosing TB infection in young children. The study’s dataset and context are valuable, but several areas need strengthening for clarity, reproducibility, and impact. A few comments to be addressed:

While the introduction describes the importance of diagnosing Mtb in children, the precise hypothesis could be more explicitly stated. Is the aim to compare test sensitivity, agreement, or predictive value for TB disease progression?

      We have re-written both the abstract and introduction to clearly state the purpose for conducting our study.  Specifically, this study was conducted to determine: 1) the prevalence of Mtb-immune sensitization in young children heavily exposed to TB using both the tuberculin skin test (TST) and QuantiFERON Gold Plus (QFT-Plus) interferon-gamma release assay, and to examine the concordance of these two tests; and 2) the diagnostic accuracy of TST and QFT-plus for confirmed and unconfirmed TB in young children. We hypothesized that in a population of young children with extensive exposure to TB, TST and IGRA testing would perform similarly to identify individuals with evidence for Mtb-sensitization and would have similar diagnostic accuracy for pediatric TB. [Lines 108-114]

The paper notes cases where TST and QFT-Plus results differ, but the clinical implications of these discordances are not deeply explored. Discussion could consider biological and operational reasons (e.g., immunosuppression, test variability).

      We have expanded the Discussion section of the manuscript to include implication of discordant results, as suggested.

The operational definition (positive TST and/or QFT-Plus) should be clearly stated upfront, and any rationale for thresholds (e.g., ≥5mm induration) should be provided.

      As detailed in the revised methods section:

                  "We considered TST positivity using two different cutoffs. First, following WHO guidelines (24), 10 mm of induration indicated a positive TST, except for CLWH or severe wasting (defined by length/height-for-weight Z-score ≤ -3) when a 5 mm cutoff was used. We derived anthropometric Z-scores from WHO child growth standards (25). Second, we applied a cutoff of 5 mm or greater of induration for all participants, regardless of HIV or nutritional status, per Center for Disease Control (CDC) guidelines for pediatric TB HHC (9)."[Lines 161-167]

      Regarding QFT-Plus based testing:

                  "QFT-Plus was performed in a CAP-certified laboratory and results interpreted according to the manufacturer’s recommendations (23). Absolute values for all four tubes (nil; mitogen; TB.1; TB.2) included in the QFT-Plus were also collected." [Lines 156-159] 

  •  
  • With 130 children, subgroup analyses (especially for TB disease vs. no disease) may be underpowered. A power calculation or discussion of statistical limitations would strengthen credibility.

We have acknowledged that our sample size is a limitation that limits statistical power in the limitations paragraph of the revised discussion. Based on prior literature, we do not advocate the use of post-hoc power calculations (Hoening and Heisey 2001, The American Statistician 55: 19-25; Goodman and Berlin 1994, Ann Intern Med 121: 200-206).  Specifically, Greenland (1988) wrote “statistical power refers to the probability of obtaining a particular type of data; it is thus not a property of particular data sets.  Statistical power of collected data…loses all meaning when one examines the result.” (Am J Epidemiol 128: 231-237). 

The applicability of findings to other high-burden or low-burden settings could be addressed more directly.

      We thank the reviewer for this suggestion and have clarified how our findings relate to high-burden and resource limited settings, in the revised conclusions.

Minor comments:

  • Ensure all tables are self-contained with clear legends, sample sizes, and units.
  • We appreciate this suggestion and have checked that all tables are all self-contained.
  •  
  • The agreement between TST and QFT-Plus could be visually summarized (e.g., Venn diagram or Bland–Altman plot).

We appreciate this suggestion and have included a Venn diagram in the revised manuscript (Figure 2).

Minor typographical errors should be corrected (e.g., “Univeristy” → “University”).

  • Thank you for identifying these errors, they have now been corrected.
  •  
  • Standardize abbreviations and ensure each is defined upon first use.
  • Thank you for these suggestions; all abbreviations have been defined upon first use.

Some key references on pediatric immune response to Mtb and the limitations of IGRAs in young children could be added.

We have added several new references related to the performance of IGRAS in the evaluation of young children for Mtb-infection and TB disease, as suggested.

  • WHO guidelines on pediatric TB diagnostics should be cited. As general comments please consider the following:

We have included the following citation: WHO consolidated guidelines on tuberculosis: Module 5: Management of tuberculosis in children and adolescents. WHO Guidelines Approved by the Guidelines Review Committee. Geneva 2022.

  •  
  • Include kappa statisticsto quantify agreement between TST and QFT-Plus.

We appreciate these suggestions and have now included Tables 3 and 4 (as well as Supplemental tables 5-7) that detail these results.

Provide sensitivity and specificity estimates using TB disease as a reference (even if imperfect).

We agree with this suggestion and have now included a new analysis of the sensitivity and specificity of the TST (5mm and 10 mm thresholds) versus QFT-Plus for pediatric TB (confirmed and unconfirmed).  Please see Table 5 of the revised manuscript.

Expand on programmatic implications: Would findings suggest one test over another for Ugandan children? Are there cost or logistical considerations?

We thank the reviewer for this suggestion and have clarified the programmatic implications of our findings in the revised conclusions.

  • Consider adding a limitations section explicitly acknowledging small sample size, potential misclassification, and absence of a gold standard for Mtb infection.

We have expanded the limitations sections of the revised manuscript to address these important concerns.

Reviewer 3 Report

Comments and Suggestions for Authors

 Gutierrez et al assessed the efficacy of TST and QFT-plus tests to detect the MTB exposure in Ugandan children with mtb sensitization. Among 130 mtb sensitized children,  TST positives (5mm cut off) are 2 times higher to develop TB and QFT-plus results are highly correlated. However, TST positives are better predictor of TB exposure than QFT-plus. Authors suggested that TST tests should be continuously performed to assess the TB exposure among mtb sensitized children.

The study reinforces the utility of TST testing in mtb sensitized children. However, there are some concerns that could be addressed.   

  1. Line 24, should be less than “5 years of age”!
  2. Line 98 to 100, although authors declared the approval of the study by both Ugandan science foundation and Cleveland medical center, no approval numbers are given. Authors should provide the approval and IRB protocol numbers.
  3. In Table 3, for QFT-plus positive test result for below 2 years, sample number is too small for statistical power. Authors should discuss it.
  4. Similarly, in suppl table 6 and 9, the QFT-plus positives for children below 2 years group had very less number, and authors should not emphasize the conclusion related to children with below 2 years of age.
  5. In suppl table 10, HIV positives have very high CI. Authors should discuss the reason for this.

Author Response

We sincerely thank the reviewers for their thoughtful and constructive comments. We greatly appreciate the time and effort invested in reviewing our manuscript. Below, we provide detailed responses to each point raised, along with corresponding revisions made to the manuscript.

Reviewer #3

Gutierrez et al assessed the efficacy of TST and QFT-plus tests to detect the MTB exposure in Ugandan children with mtb sensitization. Among 130 mtb sensitized children,  TST positives (5mm cut off) are 2 times higher to develop TB and QFT-plus results are highly correlated. However, TST positives are better predictor of TB exposure than QFT-plus. Authors suggested that TST tests should be continuously performed to assess the TB exposure among mtb sensitized children.

The study reinforces the utility of TST testing in mtb sensitized children. However, there are some concerns that could be addressed.   

  1. Line 24, should be less than “5 years of age”!

      We thank the reviewer for noting this typo that has now been corrected.

  1. Line 98 to 100, although authors declared the approval of the study by both Ugandan science foundation and Cleveland medical center, no approval numbers are given. Authors should provide the approval and IRB protocol numbers.

      We have added ethical approval numbers to the revised manuscript as suggested.

In Table 3, for QFT-plus positive test result for below 2 years, sample number is too small for statistical power. Authors should discuss it.

      We agree with the review, that the n= 4 for children under 2 years old with TB and a positive QFT-Plus results limits the strength of the statistical approach (Fisher exact test was applied here) and the generalizability of our findings. These limitations have been emphasized in the revised manuscript.

Similarly, in suppl table 6 and 9, the QFT-plus positives for children below 2 years group had very less number, and authors should not emphasize the conclusion related to children with below 2 years of age.

We agree with the review, that the n= 4 for children under 2 years old with TB and a positive QFT-Plus results limits the strength of the statistical approach (Fisher exact test was applied here) and the generalizability of our findings. These limitations have been emphasized in the revised manuscript.

In suppl table 10, HIV positives have very high CI. Authors should discuss the reason for this.

      This limitation has now been added to the revised limitations paragraph.

Round 2

Reviewer 1 Report

Comments and Suggestions for Authors

The authors have fully answered the reviewer's questions and taken into account all his comments.
The reviewer believes that the manuscript in this form is suitable for publication.

Reviewer 2 Report

Comments and Suggestions for Authors

All my comments have been succesfully addressed. I have no further comments.